# Clustered synapses develop in distinct dendritic domains in visual cortex before eye opening

Alexandra H Leighton[1], Juliette E Cheyne[1†], Christian Lohmann[1,2]*

[1]Department of Synapse and Network Development, Netherlands Institute for Neuroscience, Amsterdam, Netherlands; [2]Department of Functional Genomics, Center for Neurogenomics and Cognitive Research, VU University Amsterdam, Amsterdam, Netherlands

**\*For correspondence:**
c.lohmann@nin.knaw.nl

**Present address:** [†]Physiology Department, Centre for Brain Research, University of Auckland, Auckland, New Zealand

**Competing interest:** The authors declare that no competing interests exist.

**Abstract** Synaptic inputs to cortical neurons are highly structured in adult sensory systems, such that neighboring synapses along dendrites are activated by similar stimuli. This organization of synaptic inputs, called synaptic clustering, is required for high-fidelity signal processing, and clustered synapses can already be observed before eye opening. However, how clustered inputs emerge during development is unknown. Here, we employed concurrent in vivo whole-cell patch-clamp and dendritic calcium imaging to map spontaneous synaptic inputs to dendrites of layer 2/3 neurons in the mouse primary visual cortex during the second postnatal week until eye opening. We found that the number of functional synapses and the frequency of transmission events increase several fold during this developmental period. At the beginning of the second postnatal week, synapses assemble specifically in confined dendritic segments, whereas other segments are devoid of synapses. By the end of the second postnatal week, just before eye opening, dendrites are almost entirely covered by domains of co-active synapses. Finally, co-activity with their neighbor synapses correlates with synaptic stabilization and potentiation. Thus, clustered synapses form in distinct functional domains presumably to equip dendrites with computational modules for high-capacity sensory processing when the eyes open.

## eLife assessment

This **important** work provides insight into the activity and spatial organization of synapses during early postnatal development in the mouse visual cortex using state-of-the-art tools to show that synapses are distributed in co-active clusters well before eye opening. The evidence supporting the claims is **convincing**, and this revised version provides additional methodological details about the experimental paradigm and image analysis.. This work is of particular interest to the field of developmental neuroscience and can also be used by computational neuroscientists studying dendritic integration.

## Introduction

Building a functional brain requires neuronal circuits to be wired up with high specificity via synapses between connecting neurons. Developing neurons select their synaptic partners based on molecular cues such as adhesion molecules and subsequently refine their connections through activity-dependent synaptic plasticity, stabilization and elimination driven by spontaneous network activity, and later, after the onset sensory function, by experience (*Cline, 2003*; *Martini et al., 2021*; *Sanes and Zipursky, 2020*). Electron microscopy revealed that excitatory synapses between neurons emerge

during the end of the first postnatal week in mouse and rat sensory cortex (*Blue and Parnavelas, 1983*; *De Felipe et al., 1997*; *Miller and Peters, 1981*; *Wildenberg et al., 2023*). The addition of new synapses peaks at the end of the second postnatal week and synapse density in the primary visual and somatosensory cortices approaches adult levels once the eyes open at postnatal day (P) 14 (*Blue and Parnavelas, 1983*; *De Felipe et al., 1997*). While these structural studies clearly showed the importance of this developmental period for adding large numbers of synaptic connections in emerging circuits, how cortical dendrites establish functional inputs has not been investigated on the level of individual synapses in vivo.

In the adult cortex, synaptic inputs to dendrites are highly organized with subcellular specificity. Recent studies demonstrated that synaptic inputs of cortical pyramidal cells are clustered, such that synapses with similar activity patterns or stimulus selectivity are located near each other along their dendrites (*Iacaruso et al., 2017*; *Ju et al., 2020*; *Kerlin et al., 2019*; *Otor et al., 2022*; *Scholl et al., 2021*; *Takahashi et al., 2012*; *Wilson et al., 2016*; *Winnubst et al., 2015*). Since pyramidal cell dendrites integrate local inputs supra-linearly (*Branco and Häusser, 2011*; *Harnett et al., 2012*; *Losonczy and Magee, 2006*; *Makara and Magee, 2013*), this arrangement of clustered inputs is thought to allow for local synaptic integration in dendritic computational subunits (*Larkum, 2022*; *Larkum and Nevian, 2008*; *Major et al., 2013*; *Tran-Van-Minh et al., 2015*). Local dendritic integration dramatically increases the computational power of neurons (*Poirazi and Mel, 2001*). Furthermore, experimentally blocking supra-linear dendritic integration disturbs sensory stimulus sensitivity and selectivity in cortical neurons (*Lavzin et al., 2012*; *Palmer et al., 2014*; *Smith et al., 2013*; *Xu et al., 2012*). Finally, learning is associated with structural plasticity of clustered synapses (*Cichon and Gan, 2015*; *Fu et al., 2012*; *Hedrick et al., 2022*; *Makino and Malinow, 2011*; *McBride et al., 2008*). Together, these studies highlight the fundamental importance of the subcellular organization of synaptic inputs along dendrites.

This raises the question of how the precise subcellular organization of synapses is achieved during development. Synaptic clustering can be observed early on in the developing visual system of tadpoles and mice (*Podgorski et al., 2021*; *Winnubst et al., 2015*), suggesting that synaptic inputs develop with some degree of specificity, but the trajectory of functional synaptic input development has been unclear. Specifically, it is unknown (1) when synaptic inputs become clustered, (2) whether synapses are formed in a clustered state or arise in a random state to become sorted into clusters later, and (3) whether synapses with similar input patterns distribute smoothly along dendrites or emerge in distinct domains.

Here, we performed in vivo whole-cell patch-clamp recordings with two-photon calcium imaging (*Chen et al., 2011*; *Helmchen et al., 1999*; *Jia et al., 2010*; *Svoboda et al., 1997*; *Takahashi et al., 2012*; *Winnubst et al., 2015*) of mouse primary visual cortex layer 2/3 neurons, allowing us to map functional synaptic inputs across dendritic arborizations in the course of the second postnatal week. We combined resonant scanning with piezo-driven focusing to image large proportions of a neuron's dendrites. The total dendritic length imaged per neuron was almost 10 times larger than in our previous study (*Winnubst et al., 2015*), allowing us to map the large-scale distribution of synapses along developing dendrites.

We found that the number of functional synapses and the frequency of transmission events at individual synapses increase rapidly before eye opening. Furthermore, we observed that clustered synaptic inputs accumulate in spatially and functionally separated domains already at the beginning of the second postnatal week. By the end of the second postnatal week, dendrites had become almost entirely covered by functional domains of co-active neighbors, most likely through local plasticity driven by coincident spontaneous network activity. Thus, already before eye opening functional domains develop in cortical dendrites, which can serve as distinct computational modules for processing visual stimuli thereafter.

## Results
### Mapping synaptic inputs onto spine and shaft synapses

To visualize the activity and spatial organization of functional excitatory synapses in the developing mouse primary visual cortex, we imaged spontaneously occurring synaptic calcium transients in apical dendrites of pyramidal cells in L2/3 using the calcium indicator GCaMP6s in neonatal mice (*Figure 1A*).

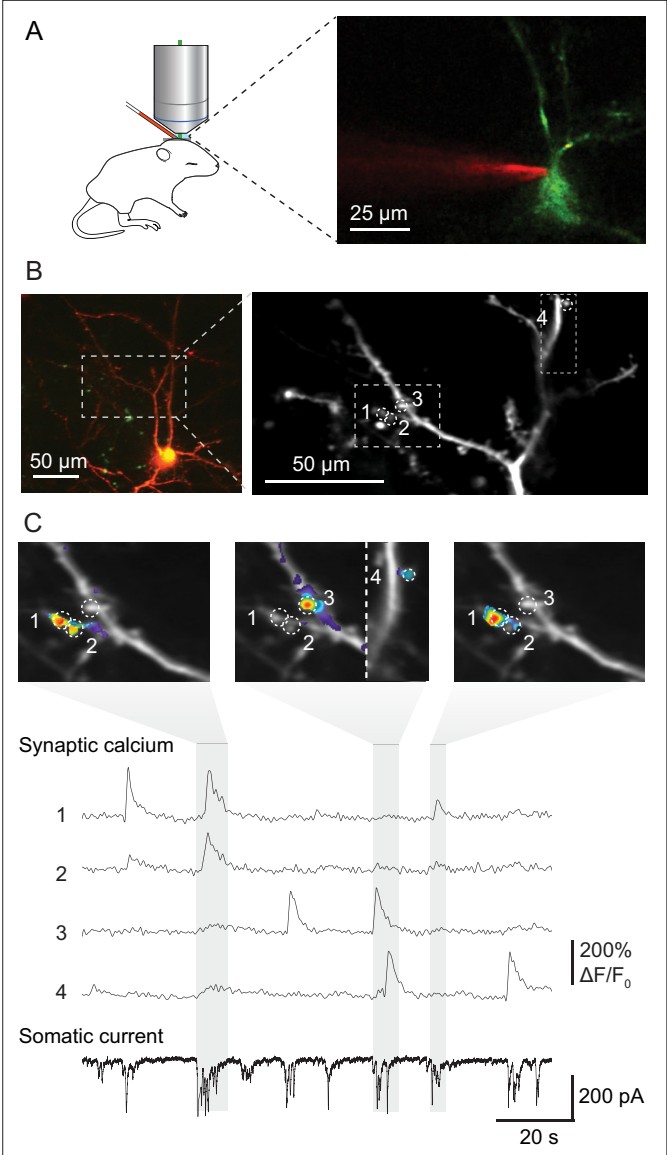

**Figure 1.** Mapping functional synaptic inputs of V1 layer 2/3 pyramidal cell dendrites in vivo. (**A**) Schematic of experimental setup (left) and pyramidal cell in mouse visual cortex expressing DsRed and GCaMP-6s (right). Layer 2/3 neurons were targeted in vivo under two-photon guidance. Patch-clamp pipettes were coated with Alexa 594 for visualization. (**B**) Left: layer 2/3 pyramidal neuron expressing DsRed and GCaMP-6s at postnatal day (P) 12 (D9 in *Figure 2—figure supplements 1 and 2*). Right: high-magnification view of the dendrite marked with dashed lines on the left. Dashed circles and numbers mark four individual synaptic sites. Dashed rectangle indicates the views shown in (**C**). (**C**) Spontaneous synaptic inputs were visible both as local increases in fluorescence at the four synaptic sites labeled in (**B**), and as synaptic currents in the somatic whole-cell voltage-clamp recording. Gray vertical bars indicate three individual barrages of synaptic inputs, and panels above show synaptic calcium increases at the four example synapses during each labeled barrage. Synaptic transmission of individual synapses could be detected and distinguished clearly from that at neighboring synapses.

We combined resonant scanning and piezo-driven z-positioning to image a large area of the dendritic tree of individual neurons (*Figure 1B*). Calcium transients at individual synapses reflecting synaptic transmission (*Jia et al., 2010*; *Takahashi et al., 2012*; *Winnubst et al., 2015*) allowed us to map synaptic inputs across 12 dendritic areas from 11 mice at ages between postnatal days 8–13, the day before eye opening. Together, these dendrites hosted 354 functional synapses, which produced 3440 synaptic transmission events. We recorded these neurons in voltage-clamp configuration to measure

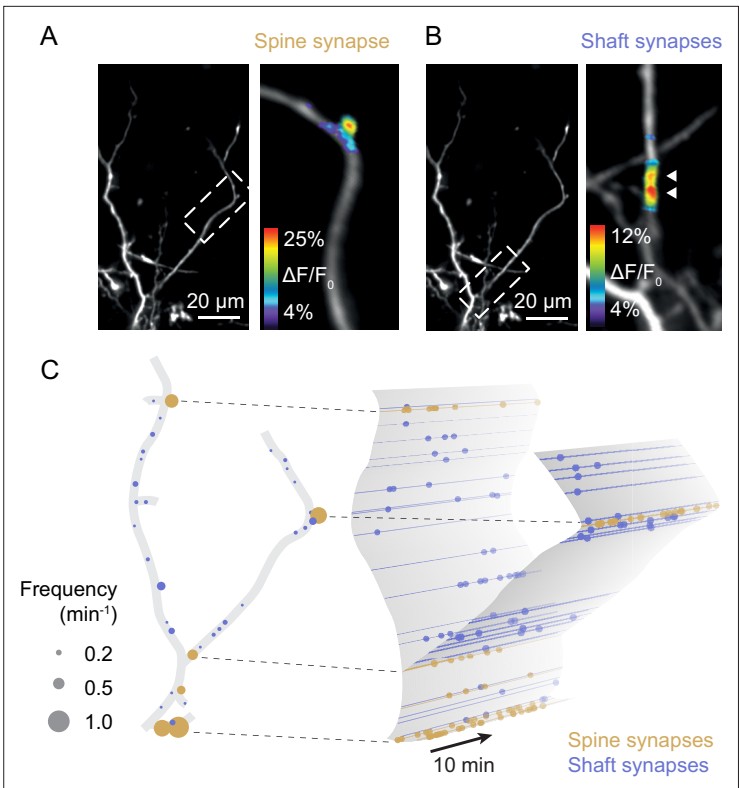

**Figure 2.** Sampling synaptic transmission in spine and shaft synapses (D3). (**A**) Synaptic transmission in a spine of a layer 2/3 pyramidal neuron at P9. (**B**) Synaptic transmission at two neighboring shaft synapses of the neuron shown in (**A**). (**C**) Schematic representation of the synaptic inputs of the neuron shown in (**A**) and (**B**). Left: all functional synapses are labeled as individual discs. Yellow and blue discs represent spine and shaft synapses, respectively. The disc diameter indicates the observed transmission frequency at each synapse. Right: schematic representation of synaptic transmission across time in the same dendrite. Dots represent individual synaptic transmission events.

The online version of this article includes the following figure supplement(s) for figure 2:

**Figure supplement 1.** Overview of all imaged dendrites and synapses.

**Figure supplement 2.** Overview of all recorded synaptic transmission events across all imaged dendrites.

**Figure supplement 3.** Identification of the patched neuron and its dendrites.

**Figure supplement 4.** Region of interest (ROI) selection.

**Figure supplement 5.** Identification of shaft and spine synapses.

barrages of synaptic currents arriving at the soma simultaneously with individual synaptic transmission events monitored by calcium imaging (*Figure 1C*).

In the mature cortex, most excitatory synapses of pyramidal neurons are located on spines (*Berry and Nedivi, 2017*; *Kasai et al., 2021*; *Moyer and Zuo, 2018*). Spines provide some chemical and electrical isolation from the dendrite's shaft and produce clearly detectable calcium signals evoked by presynaptic inputs (*Figure 2A*). However, during development, many cortical synapses are formed directly onto the shaft (*Miller and Peters, 1981*; *Wildenberg et al., 2023*), where synaptic calcium transients are masked by calcium influx triggered by back-propagating action potentials. Therefore, we prevented action potential firing by recording layer 2/3 pyramidal neurons in voltage-clamp mode during imaging and the intracellular sodium channel blocker QX314 in the patch-pipette. The holding potential was set to –30 mV to facilitate calcium influx through NMDA receptors (*Jia et al., 2010*; *Takahashi et al., 2012*; *Winnubst et al., 2015*). In this configuration, we could image local calcium transients at both spine and shaft synapses (*Figure 2A and B*, see 'Methods' for details and potential caveats) and map all functional excitatory synaptic inputs onto these dendrites across space and time (*Figure 2C*, *Figure 2—figure supplements 1–5*).

## Synaptic inputs increase rapidly during the second postnatal week

Previous electron microscopy studies revealed that synapses form at the highest rate during the second postnatal week in rodent sensory cortex (*Blue and Parnavelas, 1983*; *De Felipe et al., 1997*; *Wildenberg et al., 2023*); however, the course of functional synapse development in vivo has not been investigated yet. In line with these previous EM studies, we report here that the density of functional synapses increased significantly between P8 and P13 (*Figure 3A and B*). The frequency of synaptic transmission at individual synapses increased as well (*Figure 3C*). Hence, the number of synaptic transmission events in a given dendrite was higher by P12/13 (n = 6 dendrites) compared to the beginning of the second postnatal week (P8–10; n = 6 dendrites; *Figure 3D*). While the percentage of spine synapses was slightly higher in older dendrites, there was no significant relationship between age and the percentage of spine synapses across the sampled dendrites (*Figure 3E*). This is in line with previous observations that spine development is maximal after P14 (summarized in *Lohmann and Kessels, 2014*).

The distribution of synaptic transmission frequencies of all synapses showed that the majority of synapses received inputs at low frequencies; however, the long tail of the distribution demonstrated that a number of synapses were very active (*Figure 3F*). In general, spine synapses were more active than shaft synapses in both young (P8–10; $0.55 \pm 0.09$ min$^{-1}$ vs. $0.16 \pm 0.03$ min$^{-1}$, mean ± SEM) and older animals (P12–13; $0.67 \pm 0.07$ min$^{-1}$ vs. $0.23 \pm 0.03$ min$^{-1}$; *Figure 3G*). While we detected synaptic transmission at the majority of spines, a fraction of structural spines remained inactive during our recordings in both younger (25 ± 20%) and older dendrites (9 ± 9% for recording durations of 29 ± 8 and 26 ± 6 min, respectively; mean ± STD), possibly representing presynaptically silent synapses (*Voronin and Cherubini, 2004*).

Next, we investigated when individual synapses were activated in relation to the barrages of spontaneous synaptic inputs neurons receive during this developmental period (*Figure 4A*). Barrages were identified as transient inward currents of at least 10 pA, which consisted of multiple synaptic currents. We found that most synaptic inputs occurred in barrages (59% in young neurons, 61% in older neurons). The rate of barrages did not change significantly with age (*Figure 4B*); however, we did observe a trend toward an increase in synaptic charge transferred per barrage (*Figure 4C*), as well as a significant increase in the number of synaptic transmission events per barrage (*Figure 4D*) during development. As expected, the number of events observed per barrage correlated highly linearly with the charge transferred during each barrage (*Figure 4E*). In addition, the amplitude of individual synaptic transmission events was higher during larger barrages (*Figure 4F*), suggesting that multiple synaptic vesicles were released simultaneously during larger barrages.

## Synapses are spatially organized along developing dendrites

In the mature mouse visual cortex, synaptic inputs are functionally clustered along layer 2/3 dendrites, such that neighboring synapses frequently have similar receptive fields (*Iacaruso et al., 2017*). During development, synaptic inputs are already clustered before eye opening (*Winnubst et al., 2015*). Therefore, we explored here the developmental trajectory of dendritic input organization in general and synaptic clustering in particular. First, we noticed – mostly in young neurons – that some dendritic segments contained several synapses, whereas other segments were essentially devoid of active synapses (*Figure 5A*). To determine whether synapse distribution along the imaged dendrites was indeed patterned rather than random, we compared the distances between each synapse and its nearest distal and proximal neighbors with distances that were estimated after randomly shuffling synaptic locations along the dendrite. We found that the observed inter-synapse distance distribution differed significantly from randomized distributions in younger dendrites (P8–P10; *Figure 5B*). In contrast, in older dendrites (P12–P13) synapse distribution was indistinguishable from randomized distributions. We concluded that in younger dendrites synapses accumulated in specific dendritic segments, and that with development the entire dendritic surface became covered by synapses more evenly.

Our finding that the distribution of transmission frequencies of individual synapses showed an extended tail (*Figure 3F*) made us wonder whether those synapses with high transmission frequencies showed any particular spatial distribution as well. We defined high-activity synapses as those in the top 20 activity percentile for each age. In line with our previous observation that spine synapses showed higher transmission rates than shaft synapses (*Figure 3G*), we found that the majority (76%) of

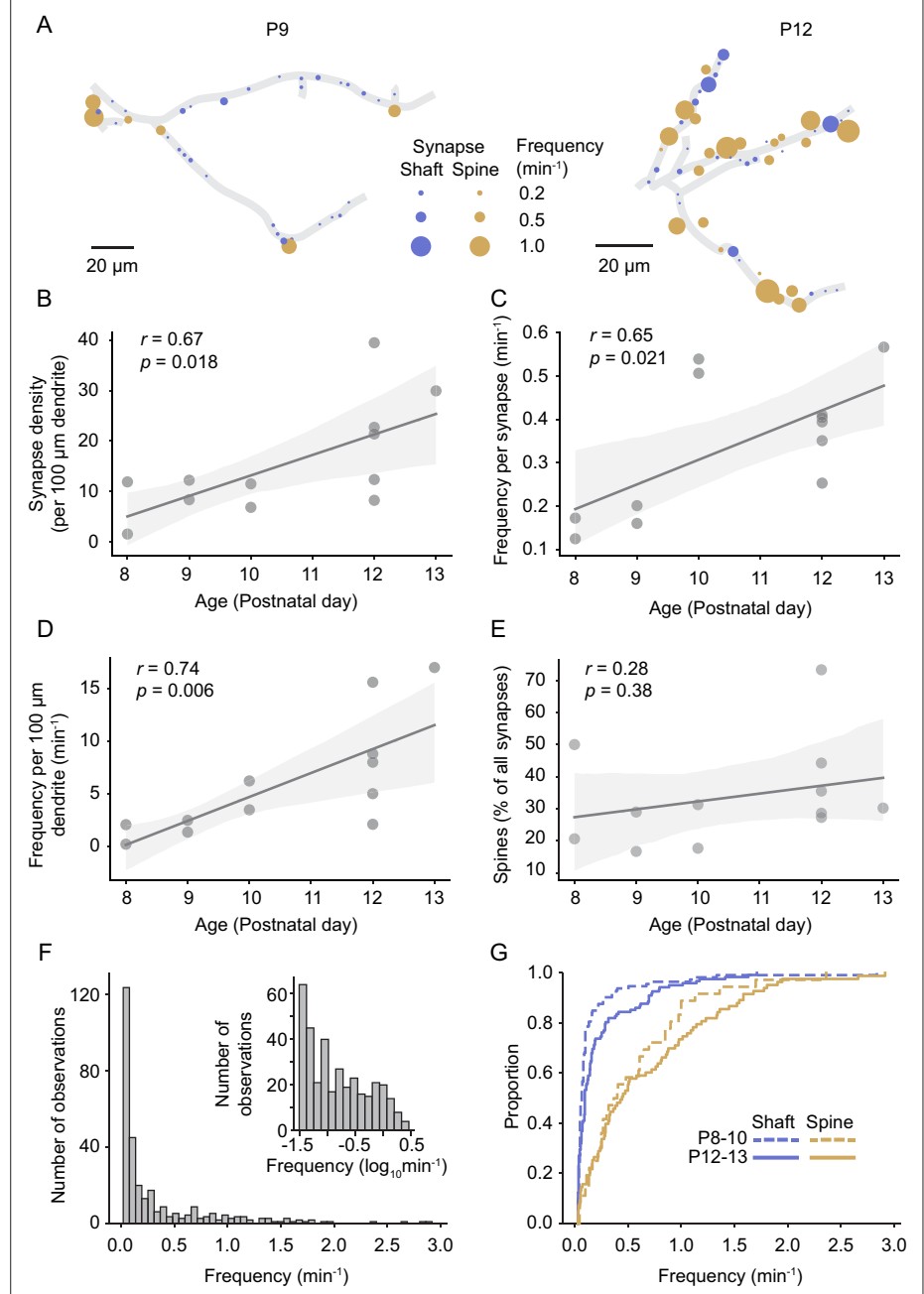

**Figure 3.** Synapse density and synaptic transmission frequency increased rapidly during the second postnatal week. (**A**) Plots of layer 2/3 pyramidal cell dendrites from two experiments at P9 (D3) and P12 (D10), respectively. Each functional synapse is indicated as a disc where the diameter represents the frequency of transmission at shaft (blue) and spine (yellow) synapses. (**B**) The density of functional synapses increased with age. Each dot represents the number of synapses per 100 μm dendrite of one experiment. Statistical parameters in this panel and (**C–E**) were derived from linear regressions on the data from n = 12 experiments. The light gray area indicates the 95% confidence interval as in all following regression plots. (**C**) The frequency of synaptic transmission per synapse increased with age. Each dot represents the mean frequency across all synapses from one experiment. (**D**) The frequency of synaptic inputs along a stretch of dendrite increased multifold during the second postnatal week. (**E**) The percentage of synapses that are located on spines did not change significantly during the observed period. (**F**) The distribution of transmission rates across all sampled synapses. While most synapses show only low transmission frequency, the distribution is very long-tailed, demonstrating that few synapses show very high synaptic transmission rates. Inset: frequency distribution shown on a logarithmic scale. (**G**) Cumulative distribution of transmission frequencies across spine and shaft synapses in younger (P8–10) and older dendrites (P12–P13). The

*Figure 3 continued on next page*

*Figure 3 continued*

transmission frequency was higher at spine than at shaft synapses at both ages (each p<10⁻⁹, Mann–Whitney *U* tests, P8–10: n = 36 spine/113 shaft synapses, N = 6 dendrites, P12–13: n = 83 spine/122 shaft synapses, N = 6 dendrites).

high-activity synapses were located on spines. Furthermore, we observed that highly active synapses occurred frequently near each other (*Figure 5C*). Therefore, we tested whether there was a relationship between the activity level of a given synapse and the distance to its nearest high-activity neighbor. We discovered that synapses that were located within 10 μm from a high-activity synapse were more active than synapses that were more distant to a high-activity synapse (*Figure 5D*). In fact, neighbors of high-activity synapses were significantly more active than those with larger distances from high-activity synapses (*Figure 5E*) and this relationship was highly consistent across all ages (*Figure 5F*).

## Synaptic inputs are organized in functional domains

Having established that high-activity synapses were often surrounded by other high-activity synapses, we defined dendritic segments that contain high-activity synapses and their neighbors as dendritic domains (*Figure 6A*). Synapses that showed normal activity levels (i.e. non-high-activity synapses) were assigned to domains when they were 10 μm or less away from a high-activity synapse. High-activity synapses that were within 10 μm of each other were assigned to the same domain, whereas those that were farther away from each other were assigned to separate domains.

Assessing domain features across development revealed that their extent increased slightly (P8–10: 18.7 ± 6.1 μm; P12–13: 22.5 ± 6.4 μm). Moreover, both the number of synapses per domain (P8–10: 3.2 ± 1.7; P12–13: 6.0 ± 3.2) as well as the density of domains along dendrites (P8–10: 1.4 ± 0.7; P12–13: 2.6 ± 1.0 per 100 μm dendrite) approximately doubled during the second postnatal week (*Figure 6B–D*). The fraction of dendritic length covered by domains had increased significantly by the end of the second postnatal week (*Figure 6E*; P8–10: 27.2 ± 16.2%; P12–13: 61.3 ± 29.4%, mean ± STD). Finally, the percentage of synapses within domains increased with age as well (P8–10: 46.2 ± 7.4%, n = 6 dendrites; P12–13: 72 ± 23.2%, n = 6 dendrites, mean ± STD; p=0.026, *t*-test).

To relate dendritic domains (i.e. segments surrounding high-activity synapses) to synapse clusters (i.e. neighboring synapses with similar activity patterns), we compared local co-activity between synapses both inside and outside of dendritic domains. For each synapse, we determined a co-activity value by dividing the number of times this synapse was co-active with any synapse in the comparison group, for example, its neighbors, by the number of times this synapse was active in total, normalized by the number of synapses in the comparison group. Next, we compared the local co-activity of synapses within domains to the local co-activity of synapses outside domains and found that domain synapses showed much higher local co-activity in both younger (in: 0.08 ± 0.02; out: 0.02 ± 0.01; mean ± SEM) and older animals (*Figure 6F*; in: 0.06 ± 0.01; out: 0.04 ± 0.01; mean ± SEM). Thus, synapses within a domain were more synchronized with their neighbors than synapses that were not part of a domain.

This finding raised the question whether dendritic domains are also functionally distinct from each other such that patterns of synaptic inputs received by one domain differed from the patterns received by the other domains on the same dendrite. To address this possibility, we quantified the co-activity of individual synapses with synapses from within their domain and compared that with their co-activity with synapses from other domains. We found that the co-activity within domains was consistently higher than the co-activity between synapses from different domains across all imaged dendrites (*Figure 6G and H*). These results demonstrate that synapses with similar input patterns emerge in distinct dendritic domains. Furthermore, they suggest that clustered synaptic inputs in the mouse visual cortex (*Iacaruso et al., 2017*; *Wilson et al., 2016*; *Winnubst et al., 2015*) are confined to developmentally emerging distinct dendritic domains and do not represent a smooth gradient of input patterns along the dendrite.

## Local synaptic synchronicity correlates with synaptic activity changes

We then asked how uniform synaptic inputs became sorted into domains. Since we had discovered previously that local co-activity drives plasticity to cluster neighboring synapses (*Niculescu et al., 2018*; *Winnubst et al., 2015*) and found here that the synapses within domains were more synchronized with

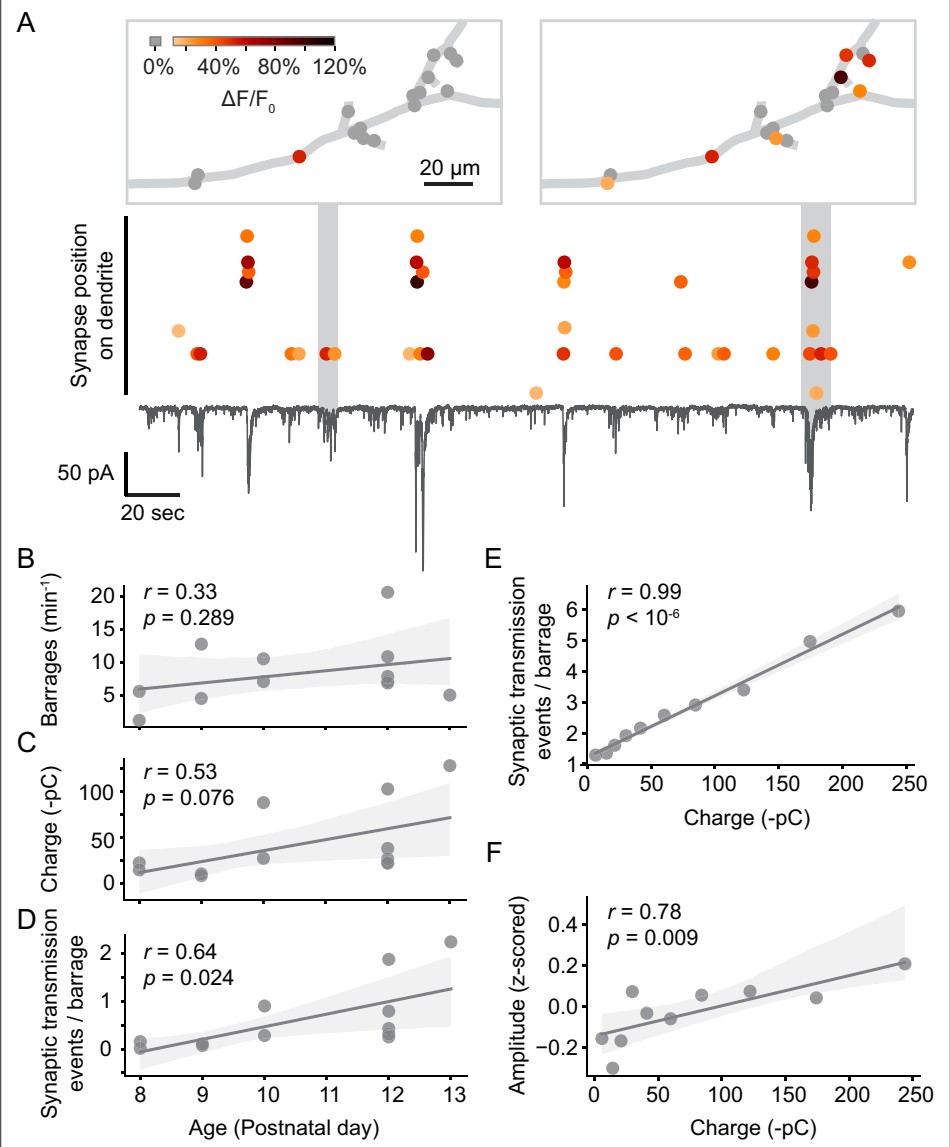

**Figure 4.** Synaptic transmission at the synapse and soma level. (**A**) Synaptic transmission events (dots) at individual synapses and synaptic currents recorded simultaneously from the soma (black trace) in a layer 2/3 neuron at P10 (D6). Top: dendritic location of synapses that are active during two example barrages. Colors represent the amplitude of individual synaptic transmission events. Gray dots show synapses that were inactive during the respective barrages. Most synaptic transmission events occurred during barrages of somatic currents. (**B**) The number of barrages did not change significantly during the observed developmental period. Each dot represents the mean from one experiment. The light gray area indicates the 95% confidence interval as in all following regression plots. (**C**) The total charge transferred during barrages of synaptic currents showed a trend toward increases with age. (**D**) The number of synaptic transmission events observed during individual barrages increased with age. (**E**) The number of synaptic transmission events correlated linearly with the total charge transferred of barrages during which they occurred. Each data point shows the mean number of synaptic transmission events per barrage for geometrically binned barrage sizes. (**F**) The amplitude of synaptic transmission events at a given synapse correlated with the total charge transferred, suggesting that more than one vesicle was released per synapse during larger barrages. Each data point shows the mean amplitude (z-scored across all events of each synapse) per barrage for geometrically binned barrage sizes.

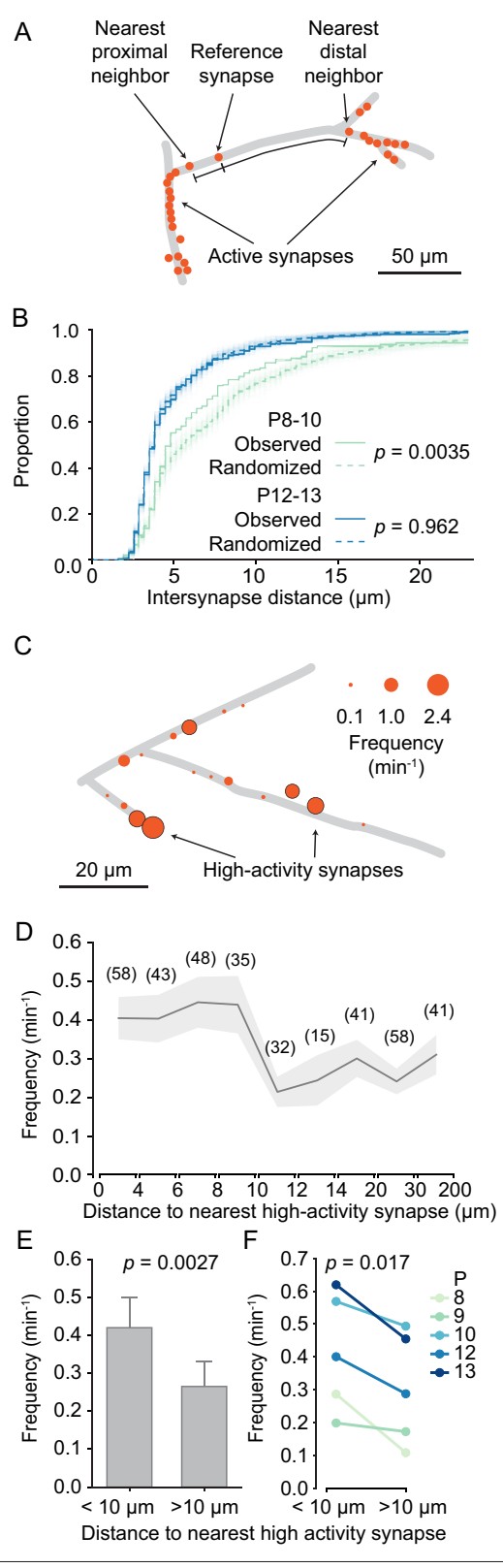

**Figure 5.** Structured organization of synapses along developing dendrites. (**A**) Graphical representation of layer 2/3 pyramidal cell dendrites at P8 (D2). Each dot shows an active synapse. Several dendritic segments carried a high density of synapses; others did not receive functional synaptic inputs. (**B**) Cumulative distribution of inter-synapse distances between each synapse and their nearest proximal and distal neighbors. The inter-synapse

*Figure 5 continued on next page*

*Figure 5 continued*

distance distribution of synapses in younger dendrites (P8–10, green, N = 6 dendrites) differed significantly from randomized distributions (100 runs are shown as faint lines and their average as dashed line). In older dendrites (P12–13, blue, N = 6 dendrites), inter-synapse distributions did not differ from randomized distributions (Kolmogorov–Smirnov tests). (**C**) Graphical representation of layer 2/3 pyramidal cell dendrites at P10 (D5). The diameter of each disc represents the transmission frequency of a synapse. Synapses that were among the 20% most active synapses within their age group (high-activity synapses) are outlined in black. Frequently, highly active synapses were located in close proximity to other highly active synapses. (**D**) The mean transmission frequency of synapses was higher at synapses that were located nearby high-activity synapses. A clear drop in transmission frequency was observed at distances larger than 10 µm from the nearest high-activity synapse. In parentheses, the number of synapses averaged for each distance bin are shown. Gray area represents SEMs. (**E**) The mean transmission frequency was higher at synapses that were located within 10 µm of a high-activity synapse than at synapses farther away from high-activity synapses (Student's *t*-test for independent samples, two-sided, n = 181 [<10 µm] and 173 [>10 µm] synapses; error bars: SEM). (**F**) Within each age group, synapses that were located within 10 µm from an high-activity synapse were more active than those that were located farther away from high-activity synapses (Student's *t*-test for paired samples, two-sided).

their neighbors than the synapses located outside domains (*Figure 6F*), we examined whether co-activity of neighboring synapses within domains predicted changes in their activity levels. To quantify changes in activity over time, we determined the Mann–Kendall score (*Hussain and Mahmud, 2019*) for each synapse. This score is positive for monotonic increases, negative for decreases, and zero in the absence of changes. Thus, synapses that increased in transmission frequency across recordings of an experiment yielded positive scores, those that underwent synaptic depression (i.e., decreased in transmission frequency) yielded negative scores, and those that maintained activity levels scored at zero (*Figure 7A*). We found that the Mann–Kendall score was more negative outside than inside domains, indicating that on average synapses outside domains tended to decrease in activity over time, whereas those inside domains showed a more positive activity trajectory (*Figure 7B*). Next, we compared each synapse's Mann–Kendall score of synaptic plasticity with its local co-activity within 10 µm proximally and distally along the dendrite and revealed a significant positive correlation (*Figure 7A and C*), demonstrating that synapses that were synchronized with their neighbors became stabilized or more active compared to those that were out of sync with their neighbors.

Together, these findings showed that visual cortex neurons established functional synapses at a very high rate during the week leading up to eye opening and suggest that developing synapses became organized into functional, spatially distinct domains based on the synchronicity with their neighbors (*Figure 7D*).

## Discussion

The fundamental idea that local processing and plasticity in dendrites is essential for high-level computations of the brain (*Mel et al., 2017*; *Poirazi et al., 2003*; *Poirazi and Mel, 2001*) has gained tremendous traction. Many recent studies show that synaptic inputs to excitatory neurons in the neocortex and hippocampus are clustered, such that synapses representing similar stimulus features are located near each other. However, when and how structured synaptic inputs assemble during development has been unclear. Here, we show that synaptic inputs to layer 2/3 pyramidal cells in the mouse visual cortex are structurally and functionally organized in distinct dendritic domains already during the second postnatal week, that is, before eye opening and detailed visual input. Furthermore, we find that local co-activity within dendritic domains correlates with synaptic activity changes, indicating that synapses are sorted into distinct functional dendritic domains through plasticity mechanisms driven by spontaneous network activity (*Figure 7D*).

We obtained insight into the spatial organization of synaptic inputs onto layer 2/3 pyramidal cell dendrites by mapping synaptic transmission during spontaneously occurring network activity in vivo. Hence, the observed synaptic activity is the result of an interaction between the firing patterns of presynaptic neurons and the features of the individual synapses on the monitored neuron. Therefore, it is important to understand which properties of synaptic inputs we can infer from the recorded input patterns. Firstly, we observed that the distribution of synaptic transmission frequencies is very long-tailed: most synapses show only rare transmission events, whereas roughly 20% of synapses are highly

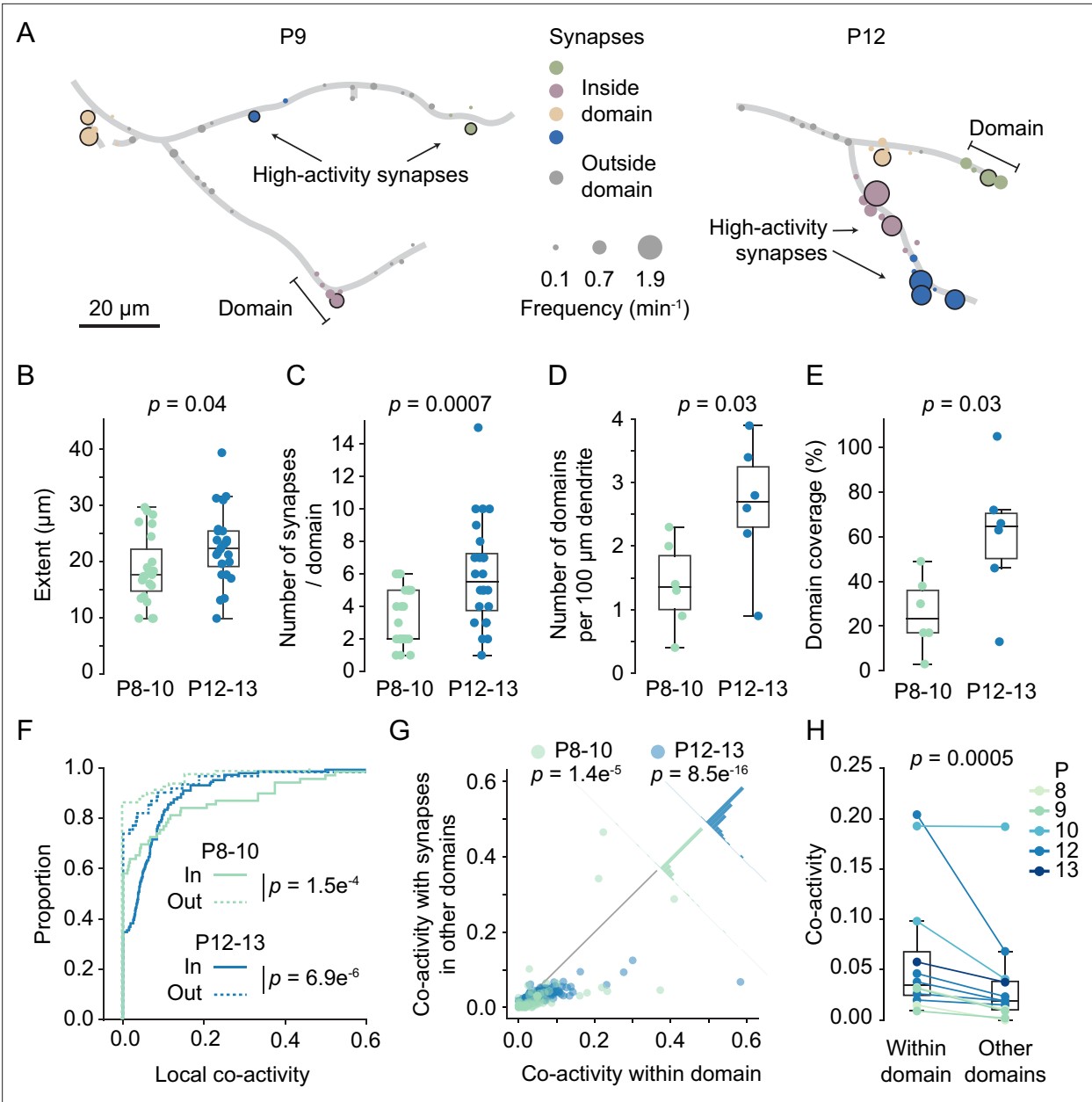

**Figure 6.** Domains of clustered synapses emerged during the second postnatal week. (**A**) Graphical representation of layer 2/3 pyramidal cell dendrites at P9 (D3) and P12 (D7). Discs represent individual synapses. The disc size indicates synaptic transmission frequency, and high-activity synapses are labeled by black outlines. Individual domains are shown in different colors. Synapses located outside domains are shown in gray. (**B, C**) The extent of domains and the number of synapses increased during development. Each dot represents one domain (P8–10: N = 6 dendrites, P12–13: N = 6 dendrites; Student's *t*-test for independent samples, two-sided, boxes indicate quartiles, and whiskers outline the entire distribution, except outliers). (**D, E**) The density of domains along dendrites and the fraction of dendrite covered by domains (**E**) increased with age. Each dot represents one dendrite (Student's *t*-test for independent samples, two-sided, boxes indicate quartiles, and whiskers outline the entire distribution, except outliers). (**F**) Cumulative distributions of local co-activity values of all recorded synapses. The local co-activity of synapses inside domains was higher than that of outside-domain synapses (Mann–Whitney *U* test, two-sided, n = 213 [in domain] and 173 [outside domain] synapses; error bars: SEM). (**G**) Synapses in domains were more co-active with their domain neighbors than with synapses in other domains. Each dot represents the co-activity of a synapse with its neighbors within the domain and its co-activity with synapses in other domains (Wilcoxon signed-rank test). The histograms perpendicular to the equality line show the distribution of differences. (**H**) For each experiment, the co-activity of synapses within domains was higher than their co-activity with synapses in other domains (Wilcoxon signed-rank test).

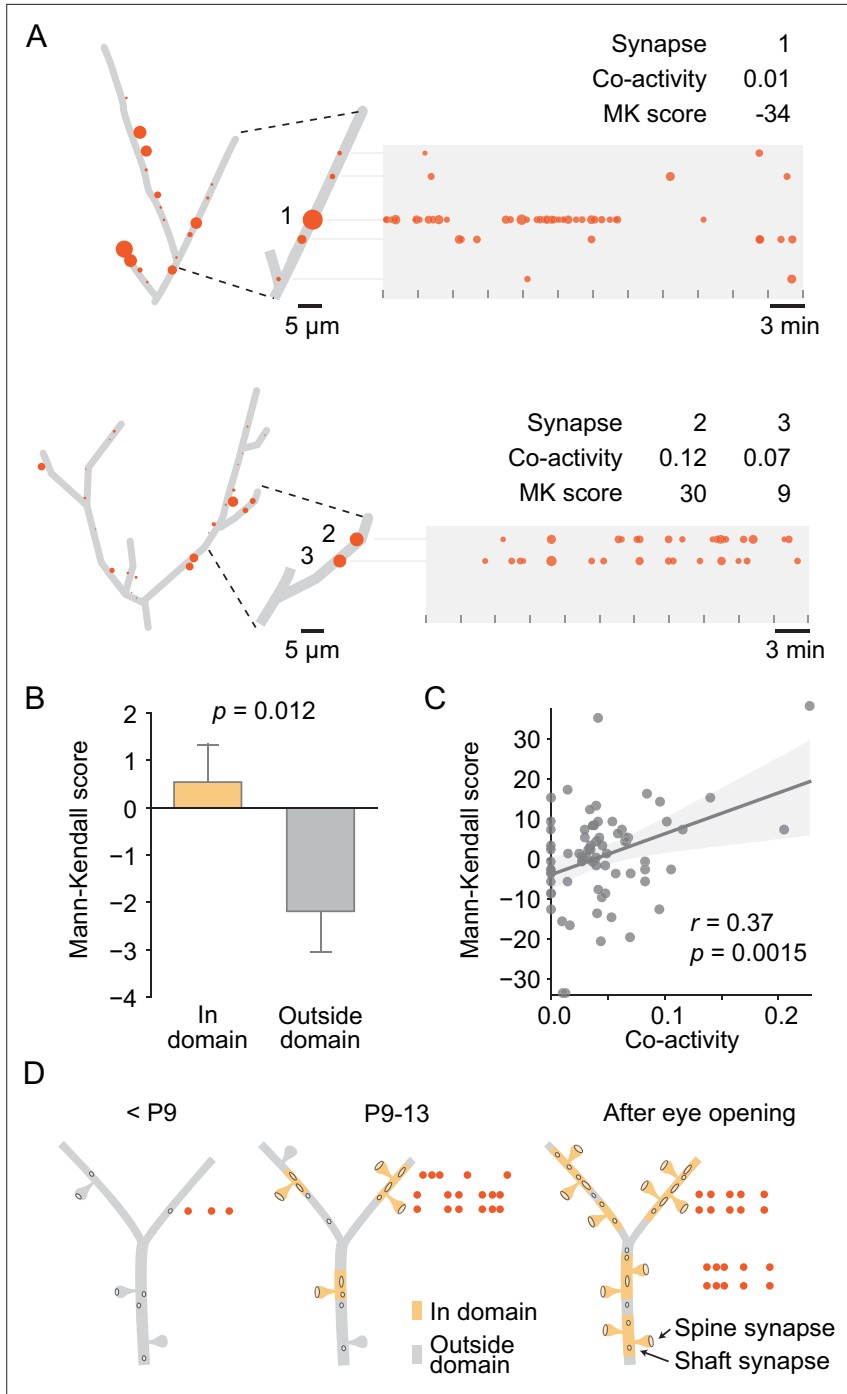

**Figure 7.** Local co-activity of synapses inside dendritic domains correlated with synaptic potentiation. (**A**) Graphical representation of two layer 2/3 pyramidal cell dendrites at P12 (D5, D9). Left: red discs on each dendrite represent active synapses, and their size shows relative activity levels. Activity of synapse (1) was mostly out of sync with its neighbors as indicated by a low co-activity score. This synapse was very active at the beginning of the experiment and stopped transmitting toward the end, resulting in a low Mann–Kendall (MK) score, which indicates a monotonic decrease in synaptic activity. Synapses (2) and (3) showed higher local co-activity and increased in activity over time as evidenced by a positive MK score. Tick marks indicate durations of individual 3 min recordings. Red dots in the time plots represent individual synaptic transmission events. The size of each data point indicates the relative amplitude of each event. (**B**) The MK score, a measure of monotonic changes in synaptic transmission frequency, was higher for domain synapses than synapses located outside domains. (Student's *t*-test for independent samples, two-sided, n = 213 [in domain] and 173 [outside domain] synapses

*Figure 7 continued on next page*

*Figure 7 continued*

from all ages; error bars: SEM). (**C**) The change in transmission frequency of a synapse was correlated with its local co-activity. Each dot represents one synapse with a transmission frequency of at least 0.6 min⁻¹ (linear regression). (**D**) Model for the establishment of domains of clustered synaptic inputs in layer 2/3 neurons of the developing visual cortex. Until P8, only a few active synapses are scattered across the developing dendrites. During the second postnatal week, synapses become sorted into distinct domains where in-sync synapses become stabilized or potentiated, and desynchronized synapses undergo synaptic depression. By eye opening, most dendrites are covered by domains of synapses that are more co-active with each other than with synapses that are located in other domains.

active. In contrast, the largest population of input neurons (i.e., other layer 2/3 pyramidal cells) show very homogeneous firing activity during neonatal spontaneous network events where all neurons participate at a similar rate (*Rochefort et al., 2009*; *Siegel et al., 2012*). Consequently, the distribution we observed here for individual synapses is most likely determined by release probability differences between individual presynaptic terminals along the dendrite. Furthermore, our current findings are congruent with our previous results, which show that the transmission frequency of individual synapses is highly plastic and regulated over a large range by local synaptic interactions and a push-pull mechanism between mature BDNF and proBDNF in pyramidal neurons (*Niculescu et al., 2018*; *Winnubst et al., 2015*). Thus, transmission frequency is a synapse feature of large dynamic range that is most likely harnessed by the developing circuit to adjust connection strength in response to fine-scale activity patterns. This conclusion is supported by a previous study reporting that differences in cortico-cortical synaptic connections are mostly based on differences in synaptic release probabilities, and that these release probabilities change in response to altered input patterns (*Markram et al., 1997*). Furthermore, the amplitude of calcium transients in axonal boutons varies over an order of magnitude, most likely representing a similar degree of variability in release probability, in juvenile rat layer 2/3 pyramidal cells (*Koester and Sakmann, 2000*).

How many independent inputs, that is, from unique presynaptic neurons, become connected to one dendritic domain? We observed a mean increase from three to six synapses per domain during the second postnatal week. In the adult, connected cortical neurons typically connect through fewer than 10 synapses, most of which are located on different dendrites (*Feldmeyer et al., 2006*; *Markram et al., 1997*; *Winfield et al., 1981*). However, axons that establish more than one synapse with individual dendritic segments have been observed as well. For example, in the mature somatosensory cortex, approximately 10% of synapses are redundant, that is, dendritic segments receive another synapse (or more) from the same axon (*Kasthuri et al., 2015*). At the ages investigated in the present study, synapse density is lower than that in the mature sensory cortex and redundant synapses most likely rarer. Nevertheless, our data set probably contains redundant synapses, too. Interestingly, full connectomic reconstruction of mature cortical dendrites and their presynaptic axons suggested that synaptic partner selection based on coincident activity may be responsible for the formation of multiple synapses of one axon with the same dendritic segment (*Kasthuri et al., 2015*), indicating that multiple innervation may be generated through the here-described plasticity mechanism that helps shape domains of co-active synapses.

In the mature visual system, V1 pyramidal cell synaptic inputs are clustered, such that similar stimulus features are represented by neighboring synapses (*Iacaruso et al., 2017*; *Wilson et al., 2016*). This input organization is thought to underlie local dendritic computations that maximize the computational power of cortical pyramidal neurons (*Poirazi and Mel, 2001*). In fact, perturbing local dendritic integration has been shown to decrease the sensitivity or selectivity of cortical neurons to stimuli of different modalities (*Lavzin et al., 2012*; *Palmer et al., 2014*; *Smith et al., 2013*; *Xu et al., 2012*). Which stimulus features may be encoded by synaptic domains established before eye opening? In the adult, neighboring synapses are more likely to share receptive field properties than more distant synapses. For example, in ferret V1 layer 2/3 neurons, synapses are clustered according to orientation preference (*Wilson et al., 2016*), whereas in mouse V1 layer 2/3 neurons, synapses are clustered based on receptive field localization within the visual field, as well as receptive field similarity, but not on orientation preference (*Iacaruso et al., 2017*). Thus, different sensory input features can be arranged in neighborhoods along dendrites, but exactly which features are clustered differs between species. In the retina, spontaneously generated activity travels in waves with specific directions as

well as wavefronts of defined orientations. In addition, neighboring neurons are frequently co-active during wave propagating (*Feller et al., 1997*; *Meister et al., 1991*). Hence, retinal waves encode retinotopy as well as direction and orientation selectivity. Since wave activity in retinal ganglion cells is transmitted along the central visual pathways from the retina to the visual cortex (*Ackman et al., 2012*; *Hanganu et al., 2006*; *Siegel et al., 2012*), retinal waves can shape dendritic domains based on neighborhood relationships of retinal ganglion cells representing location in visual space, and orientation and direction selectivity (*Kirchner and Gjorgjieva, 2021*). Finally, our observation that functional synapses form in distinct domains along dendrites, whereas other dendritic segments are devoid of synapses at the beginning of the second postnatal week, indicates that synapses cluster in defined domains rather than distributing evenly along dendrites.

Together, patterned spontaneous activity can be sufficient to sort synaptic inputs into domains to establish discrete computational modules along dendrites to prepare the visual cortex for processing with high sensitivity and selectivity before the eyes open. This 'first guess' will be refined further by visual inputs after eye opening, based on the statistics of sensory inputs in the actual environment.

## Methods

### Animals

All experimental procedures were approved by the Institutional Animal Care and Use Committee of the Royal Netherlands Academy of Arts and Sciences. We used 11 C57BL/6J mouse pups between P8 and P13.

### Plasmids

For in utero electroporation, GCaMP6s (Addgene plasmid 40753; Douglas Kim; *Chen et al., 2013*) was cloned into pCAGGS and used in combination with DsRed Express in pCAGGS (*Winnubst et al., 2015*, gift from Christiaan Levelt). Cells co-expressed the fluorescent protein DsRed for somatic targeting and structural information.

### In utero electroporation

Constructs were introduced through in utero electroporation at E16.5. Pyramidal neurons in layer 2/3 of the visual cortex were transfected with GCaMP6s (2 mg/ml) and DsRed (0.5–2 mg/ml) at E16.5 using in utero electroporation (*Winnubst et al., 2015*). Pregnant mice were anesthetized with isoflurane and a small incision (1.5–2 cm) was made in the abdominal wall. The uterine horns were removed from the abdomen, and DNA (1 µl) was injected into the lateral ventricle of embryos using a sharp glass pipette. Voltage pulses (five square wave pulses, 30 V, 50 ms duration, 950 ms interval, custom-built electroporator) were delivered across the brain with tweezer electrodes covered in conductive gel. Uterine horns were rinsed with warm saline solution and carefully returned to the abdomen, after which the muscle and skin were sutured.

### In vivo electrophysiology and calcium imaging

The surgery and stabilization for the in vivo calcium imaging experiments were performed as described previously (*Siegel et al., 2012*; *Winnubst et al., 2015*). Animals were anesthetized with 2% isoflurane, which was reduced to 0.7–1% after surgery. We previously reported that although this low level of anesthesia does reduce the frequency of spontaneous network events, relative to that seen in awake animals, it does not change the basic properties of spontaneous network activity, such as participation rates and event amplitudes (*Siegel et al., 2012*). Furthermore, we found previously that synaptic inputs are organized very similarly in vivo under light anesthesia and in slice cultures (*Winnubst et al., 2015*). Together, these observations indicate that the activity patterns investigated here are not or only slightly affected by low-level anesthesia. Glass electrodes (4.5–6 MΩ) were fluorescently coated with BSA-Alexa 594 to allow targeted whole-cell recordings (*Sasaki et al., 2012*) of layer 2/3 neurons located in V1, based on published coordinates (*Paxinos et al., 1991*) and our previous experience (*Leighton et al., 2021*). To visualize synaptic inputs, action potentials were blocked with QX314 in the intracellular solution (120 mM CsMeSO$_3$, 8 mM NaCl, 15 mM CsCl$_2$, 10 mM TEA-Cl, 10 mM HEPES, 5 mM QX-314 bromide, 4 mM MgATP, and 0.3 mM Na-GTP; *Takahashi et al., 2012*; *Winnubst et al., 2015*). Currents were recorded in voltage-clamp mode at 10 kHz and filtered at 3 kHz (Multiclamp

700b; Molecular Devices). To facilitate the detection of synaptic calcium transients, neurons were depolarized to –30 mV to increase NMDA receptor activation (*Takahashi et al., 2012*; *Winnubst et al., 2015*). When switching to –30 mV, the baseline fluorescence of the patched neurons increased, but not of the neighboring neurons, allowing unequivocal identification of all dendrites in the imaged volume that belonged to the patched neuron (*Figure 2—figure supplement 3*). No correction was made for the liquid junction potential. Barrages were identified as fast-rising inward currents of at least 10 pA, followed by a return to baseline, which consisted of multiple synaptic currents and were separated by at least 3 s.

## Image acquisition

In vivo calcium imaging was performed on a Nikon (A1R-MP) with a 0.8/16× water-immersion objective and a Ti:Sapphire laser (Chameleon II, Coherent). Dendrites were imaged at 5–15 Hz with a pixel size of 0.32 µm in single planes or small stacks with planes spaced 1.5–2 µm. We recorded the movement signal of the scan mirrors to synchronize calcium imaging and electrophysiology.

## Image processing

To remove drift and movement artifacts from each recording, we performed drift correction using NoRMCorre (*Pnevmatikakis and Giovannucci, 2017*). Each recording was aligned to the first recording in the series to remove any movements between recording sessions. Delta-F stacks were made using the average fluorescence per pixel as baseline. Local synaptic calcium transients were clearly detectable in the delta-F stacks.

Regions of interest (ROIs), representing putative synapses, were hand-drawn using ImageJ (NIH) at sites where spines were visible and at clearly observed activity sites (*Figure 2—figure supplement 4*). Finally, all remaining dendritic segments were filled with ROIs to ensure that no sites were missed. Putative synaptic transmission events were automatically identified across all ROIs as mean fluorescence increases that were at least twofold higher than the noise level. These putative signals were then manually reviewed using the full image recordings. Signals were rejected if they were shorter than two frames or coincided with movement. Furthermore, we ensured that there was a clearly visible local maximum at the location of the synapse and that it was stable for the duration of the signal. Automated transient detection and further data processing were performed using custom-made MATLAB (MathWorks) and Python software (Python Software Foundation; *Leighton et al., 2023*).

## Identification of spine and shaft synapses

We identified spine and shaft synapses as follows. Synaptic calcium transients that occurred in small appendages from the dendrite were defined as transmission events at spine synapses (e.g., *Figure 2A*, *Figure 2—figure supplement 5A*). Synaptic calcium transients located in the center of the dendrite, but at sites that showed small, sharply confined areas of increased fluorescence and where 3D reconstructions suggested that a spine may be located above or below the dendrite in the z-dimension were defined as spine transmission events as well (e.g., *Figure 2—figure supplement 5B*). Synaptic calcium transients that occurred in the center of 'smooth' dendritic stretches (without clearly defined spots of increased brightness) were identified as transmission events at shaft synapses (e.g., *Figure 2B*, *Figure 2—figure supplement 5A*). Since the z-resolution of two-photon microscopy is lower in the z-dimension than in xy, we may have misidentified spine synapses as shaft synapses in a few cases.

## Statistical analysis

One dendrite was imaged from each animal except once, when two dendrites were imaged in a single cell in a P12 animal. We used only synaptic sites that were active with a transmission frequency of at least 0.03 per minute. We tested whether the localization of functional synapses was structured or random along the dendrite. To establish randomized distributions of synapse positions, we first assigned 'empty' positions between the positions of actual functional synapses such that the entire imaged dendrite was covered with real or empty positions at the typical minimal inter-synapse distance (4 ± 1 µm). Next we determined all pair-wise distances between all synapses and all empty locations along dendrites using the Shortest Path with Obstacle Avoidance function of MATLAB. Then we shuffled all functional synapses across the real and empty positions randomly. For each shuffle and

the observed localizations, we determined the distances of each synapse to its nearest proximal and distal neighbors. Finally, we compared the results of 100 shuffles with the observed distribution using a Kolmogorov–Smirnov test.

The co-activity of a given synapse (e.g., with its neighbors within 10 μm, with its neighbors in the same domain or with synapses in other domains) was determined by dividing the number of transmission events at the observed synapse by the number of transmission events of the comparison group (e.g., its neighbors) during the same barrages when the observed synapse was active. This number was divided by the number of synapses in the comparison group to normalize to the size of the comparison group. We defined co-activity based on co-transmission during the same barrage because we had previously discovered that co-activity during barrages is crucial for local synaptic plasticity (*Winnubst et al., 2015*).

Domains were defined as dendritic segments that contained at least one high-activity synapse (see 'Results' for definition). If the distance between two high-activity synapses was 10 μm or less, both were assigned to the same domain. Normal synapses (those that were not high-activity synapses) were assigned to a domain if their distance to the nearest high-activity synapse was 10 μm or less. Normal synapses located between two high-activity synapses from distinct domains with 20 μm or less distance were assigned to the domain of the closer high-activity synapses. The extent of dendritic domains was determined as the distance between its most distant members. The length was padded at the edges with 5 μm from a high-activity synapse if there was no normal member within this distance. Thus single high-activity synapse domains had a length of 10 μm.

For normally distributed data and data with relatively low sample numbers (6–8), where normality tests are underpowered, we used paired and unpaired, two-tailed Student's *t*-tests as recommended (*de Winter, 2013*). For data that were not normally distributed, we used nonparametric statistical comparisons, Mann–Whitney *U* tests for independent samples and Wilcoxon signed-rank tests for paired data. To test whether investigated parameters changed with age, we performed linear regression analyses. To test the differences between distributions, we used a Kolmogorov–Smirnov test.

## Acknowledgements

We thank Jan Kirchner and Julijana Gjorgjieva for discussions, Christiaan Levelt for sharing plasmids, and Helmut Kessels, Alexander Heimel, Wei Wei, David Cabrera, Tamara Buijs, and Julijana Gjorgjieva for critically reading this article. This work was supported by grants of the Netherlands Organization for Scientific Research (NWO, ALW Open Program grants, no. 819.02.017, 822.02.006 and ALWOP.216; ENW Open Competition grant no. OCENW.KLEIN.535, ALW Vici, no. 865.12.001), ZonMW (top grant no. 9126021), and the 'Stichting Vrienden van het Herseninstituut' (all CL).

## Additional information

### Funding

| Funder | Grant reference number | Author |
| --- | --- | --- |
| Nederlandse Organisatie voor Wetenschappelijk Onderzoek | 819.02.017 | Christian Lohmann |
| Nederlandse Organisatie voor Wetenschappelijk Onderzoek | 822.02.006 | Christian Lohmann |
| Nederlandse Organisatie voor Wetenschappelijk Onderzoek | ALWOP.216 | Christian Lohmann |
| Nederlandse Organisatie voor Wetenschappelijk Onderzoek | OCENW.KLEIN.535 | Christian Lohmann |

| Funder | Grant reference number | Author |
|---|---|---|
| Nederlandse Organisatie voor Wetenschappelijk Onderzoek | 865.12.001 | Christian Lohmann |
| ZonMw | 9126021 | Christian Lohmann |
| Nederlandse Organisatie voor Wetenschappelijk Onderzoek | OCENW.M.22.310 | Christian Lohmann |
| Stichting Vrienden van het Herseninstituut | 805254845 | Christian Lohmann |

The funders had no role in study design, data collection and interpretation, or the decision to submit the work for publication.

### Author contributions

Alexandra H Leighton, Conceptualization, Formal analysis, Investigation, Writing - original draft, Writing – review and editing; Juliette E Cheyne, Conceptualization, Formal analysis, Investigation, Writing – review and editing; Christian Lohmann, Conceptualization, Formal analysis, Writing - original draft, Writing – review and editing

### Author ORCIDs

Christian Lohmann (iD) http://orcid.org/0000-0002-1780-2419

### Ethics

All experiments were approved by the institutional animal care and use committee of the Royal Netherlands Academy of Arts and Sciences under Central Committee Animal experiments (CCD) license AVD-801002015249.

Reviewer #1 (Public review): https://doi.org/10.7554/eLife.93498.3.sa1
Reviewer #2 (Public review): https://doi.org/10.7554/eLife.93498.3.sa2
Reviewer #3 (Public review): https://doi.org/10.7554/eLife.93498.3.sa3
Author response https://doi.org/10.7554/eLife.93498.3.sa4

## Additional files

### Supplementary files

• MDAR checklist

### Data availability

All data and code to reproduce the figures can be found at: https://figshare.com/s/3321a023e1da75a15676.

The following dataset was generated:

| Author(s) | Year | Dataset title | Dataset URL | Database and Identifier |
|---|---|---|---|---|
| Leighton AH, Cheyne JE, Lohmann C | 2023 | Dendritic domain development | https://doi.org/10.6084/m9.figshare.22674748.v3 | figshare, 10.6084/m9.figshare.22674748.v3 |

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

## Appendix 1

**Appendix 1—key resources table**

| Reagent type (species) or resource | Designation | Source or reference | Identifiers | Additional information |
|---|---|---|---|---|
| Strain (*Mus musculus*, both sexes, postnatal days 8–13) | C57BL/6J | Janvier | SC-C57J-F | |
| Recombinant DNA reagent | GCaMP6s, cloned into pCAGGS (plasmid) | Douglas Kim; *Chen et al., 2013* | Addgene 40753 | Calcium indicator |
| Recombinant DNA reagent | DsRed, cloned into pCAGGS (plasmid) | Christiaan Levelt | | |
| Software, algorithm | MATLAB scripts | This paper; *Leighton et al., 2023* | https://figshare.com/articles/dataset/Domain_development/22674748 | |
| Software, algorithm | Jupyter Notebooks, Python | This paper; *Leighton et al., 2023* | https://figshare.com/articles/dataset/Domain_development/22674748 | |

