## [Editor Report · eLife assessment]

This **important** work provides insight into the activity and spatial organization of synapses during early postnatal development in the mouse visual cortex using state-of-the-art tools to show that synapses are distributed in co-active clusters well before eye opening. The evidence supporting the claims is **convincing**, and this revised version provides additional methodological details about the experimental paradigm and image analysis.. This work is of particular interest to the field of developmental neuroscience and can also be used by computational neuroscientists studying dendritic integration.

---

## [Referee Report · Reviewer #1 (Public review)]

Summary:

Using concurrent in vivo whole-cell patch clamp and dendritic calcium imaging, the authors characterized how functional synaptic inputs across dendritic arborizations of mouse primary visual cortex layer 2/3 neurons emerge during the second postnatal week. They were able to identify spatially and functionally separated domains of clustered synapses in these neurons even before eye-opening and characterize how the clustering changes from P8 to P13.

Strengths:

The work is technically challenging and the findings are novel. The results support previous EM and immunostaining studies but really provide in vivo evidence on the time course and the trajectory of how functional synaptic input develop.

Weaknesses:

The authors have provided additional details about the analyses and have adequately addressed all my concerns.

---

## [Referee Report · Reviewer #2 (Public review)]

In this study, Leighton et al performed remarkable experiments by combining in-vivo patch-clamp recording with two-photon dendritic Ca2+ imaging. The voltage-clamp mode is a major improvement over the pioneer versions of this combinatorial experiment that had led to major breakthroughs in the neuroscience field for visualizing and understanding synaptic input activities in single cells in-vivo (sharp electrodes: Svoboda et al, Nature 1997, Helmchen et al, Nature Neurosci 1999; whole-cell current-clamp: Jia et al, Nature 2010, Chen et al, Nature 2011. I suggest that these papers would be cited). This is because in voltage-clamp mode, despite a full control of membrane voltage in-vivo is not realistic, is nevertheless most effective in preventing back-propagation action potentials, which would severely confound the measurement of individual synaptically-induced Ca2+ influx events. Furthermore, clamping the cell body at a strongly depolarized potential (here the authors did -30mV) also facilitates the detection of synaptically-induced Ca2+ influx. As a result, the authors successfully recorded high-quality Ca2+ imaging data that can be used for precise analysis. To date, even in view of the rapid progress of voltage-sensitive indicators and relevant imaging technologies in the recent years, this very old 'art' of combining single-cell electrophysiology and two-photon imaging (ordinary, raster-scanned, video-rate imaging) of Ca2+ signals still enable measurements of the best-level precision.

On the other hand, the interpretation of data in this study is a bit narrow-minded and lacks a comprehensive picture. Some suggestions to improve the manuscript are as follows:

(1) The authors made a segregation of 'spine synapse' and 'shaft synapse' based solely on the two-photon images in-vivo. However, caution shall be taken here, because the optical resolution under in-vivo imaging conditions like this cannot reliably tell apart whether a bright spot within or partially overlapping a segment of dendrite is a spine on top (or below) of it. Therefore, what the authors consider as a 'shaft synapse' (by detecting Ca2+ hotspots) has an unknown probability to be just a spine on top or below the dendrite. If there is other imaging data of higher axial resolution to validate or calibrate, the authors shall take some further considerations or analysis to check the consistency of their data, as the authors do need such a segregation between spine and shaft synapses to show how they evolve over the brain development stages.

(2) The use of terminology 'bursts of spontaneous inputs' for describing voltage-clamp data seems improper. Conventionally, 'burst' refers to suprathreshold spike firing events, but here, the authors use 'burst' to refer to inward synaptic currents collected at the cell body. It is obvious that not every excitatory synaptic input (or ensemble of inputs) activation will lead to spike firing under naturalistic conditions, therefore, these two concepts are not equivalent. It is recommended to use 'barrage of inputs' instead of 'burst of inputs'. Imagine a full picture of the entire dendritic tree, the fact that the authors could always capture spontaneous Ca2+ events here and there within a few pieces of dendrites within an arbitrary field-of-view suggest that, the whole dendritic tree must have many more such events going on as a barrage while the author's patch electrode picks up the summed current flow from the whole dendritic tree.

(3) Following the above issue, an analysis of the temporal correlation between synaptic (not segregating 'spine' or 'shaft') Ca2+ events and EPSCs is absent. Again, the authors drew arbitrary time windows to clump the events for statistical analysis. However, the demonstrated example data already show that the onset times of individual synaptic Ca2+ events do not necessarily align with the beginning of a 'barrage' inward current event.

(4) The authors claim that "these observations indicate that the activity patterns investigated here are not or only slightly affected by low-level anesthesia". It would be nice to show some of the recordings in this work without any anesthesia to support this claim.

(5) I suggest the authors should provide the number of cells and mice recorded in the figure legends.

(6) Instead of showing only cartoon illustrations of dendrites in Figure 3-6, I suggest showing the two-photon images as well together with the cartoon.

The authors have addressed most of my issues, but I miss the responses to my points 5 and 6. I have no additional comments.

---

## [Referee Report · Reviewer #3 (Public review)]

Summary:

There is a growing body of literature on the clustering of co-active synapses in adult mice, which has important implications for understanding dendritic integration and sensory processing more broadly. However, it has been unclear when this spatial organization of co-active synapses arises during development. In this manuscript, Leighton et al. investigate the emergence of spatially organized, co-active synapses on pyramidal dendrites in the mouse visual cortex before eye opening. They find that some dendrite segments contain highly active synapses that are co-active with their neighbors as early as postnatal day (P) 8-10, and that these domains of co-active synapses increase their coverage of the dendritic arbor by P12-13. Interestingly, Leighton et al. demonstrate that synapses co-active with their neighbors are more likely to increase their activity across a single recording session, compared to synapses that are not co-active with their neighbors, suggesting local plasticity driven by coincident activity before eye opening.

The current manuscript includes some replication of earlier results from the same research group (Winnubst et al., 2015), including the presence of clustered, co-active synapses in the visual cortex of mouse pups, and the finding that synapses co-active with their neighbors show an increase in transmission frequency during a recording session. The main novelty in the current study compared to Winnubst et al. (2015) is the inclusion of younger animals (P8-13 in the current study compared to P10-15 in Winnubst et al., 2015). The current manuscript is the first demonstration that active synapses are clustered on specific dendrite segments as early as P8-10 in the mouse visual cortex, and the first to show the progression in active synapse distribution along the dendrite during the 2nd postnatal week. These results from visual cortex may help inform our understanding of sensory development more broadly.

Strengths:

The authors ask a novel question about the emergence of synaptic spatial organization, and they use well-chosen techniques that directly address their questions despite the challenging nature of these techniques. To capture both structural and functional information from dendrites simultaneously, the authors performed whole-cell voltage clamp to record synaptic currents arriving at the soma while imaging calcium influx at individual synaptic sites on dendrites. The simultaneous voltage clamp and calcium imaging allowed the authors to isolate individual synaptic inputs without their occlusion by widespread calcium influx from back-propagating action potentials. Achieving in vivo dendrite imaging in live mice that are as young as P8 is challenging, and the resulting data provides a unique view of synaptic activity along individual dendrites in the visual cortex at an early stage in development that is otherwise difficult to assess.

The authors provide convincing evidence that synapses are more likely to be co-active with their neighbors compared to synapses located farther away (Fig. 6F-H), and that synapses co-active with their neighbors increase their transmission frequency during a recording session (Fig. 7C). These findings are particularly interesting given that the recordings occur before eye opening, suggesting a relationship between co-activity and local synaptic plasticity even before the onset of detailed visual input. These results replicate previously published findings from P10-15 pups (Winnubst et al., 2015), increasing confidence in the reproducibility of the data.

The authors also provide novel data documenting for the first time spatially organized, co-active synapses in pups as young as P8. Comparing the younger (P8-10) and older (P12-13) pups, provides insight into how clusters of co-active synapses might emerge during development.

Weaknesses:

The P8-10 vs P12-13 age comparisons are the primary novel finding in this manuscript, and it is therefore critical to avoid systematic age differences in the methods and analysis whenever possible. In their rebuttal and revised manuscript the authors have acceptably addressed prior concerns regarding this important point, as well as most of the other methodological issues raised.

One point addressed in the rebuttal, but not corrected in the manuscript relates to the reliable localization of cells to visual cortex.

---

## [Author Response]

The following is the authors’ response to the original reviews.

**Public Reviews:**

**Reviewer #1 (Public Review):**
Summary:Using concurrent in vivo whole-cell patch clamp and dendritic calcium imaging, the authors characterized how functional synaptic inputs across dendritic arborizations of mouse primary visual cortex layer 2/3 neurons emerge during the second postnatal week. They were able to identify spatially and functionally separated domains of clustered synapses in these neurons even before eye-opening and characterize how the clustering changes from P8 to P13.Strengths:The work is technically challenging and the findings are novel. The results support previous EM and immunostaining studies but provide in vivo evidence on the time course and the trajectory of how functional synaptic input develops.Weaknesses:There are some missing details about how the experiments were performed, and I also have some questions about the analyses.

We have now added a more detailed description of the methods and added new supplemental figures and descriptions to clarify our analyses. Please find our responses to the specific points of this reviewer in the section “Recommendations for the authors” below.

**Reviewer #2 (Public Review):**
In this study, Leighton et al performed remarkable experiments by combining in-vivo patch-clamp recording with two-photon dendritic Ca2+ imaging. The voltage-clamp mode is a major improvement over the pioneer versions of this combinatorial experiment that has led to major breakthroughs in the neuroscience field for visualizing and understanding synaptic input activities in single cells in-vivo (sharp electrodes: Svoboda et al, Nature 1997, Helmchen et al, Nature Neurosci 1999; whole-cell current-clamp: Jia et al, Nature 2010, Chen et al, Nature 2011. I suggest that these papers would be cited). This is because in voltage-clamp mode, despite the full control of membrane voltage in-vivo not being realistic, is nevertheless most effective in preventing back-propagation action potentials, which would severely confound the measurement of individual synaptically-induced Ca2+ influx events. Furthermore, clamping the cell body at a strongly depolarized potential (here the authors did -30mV) also facilitates the detection of synaptically-induced Ca2+ influx. As a result, the authors successfully recorded high-quality Ca2+ imaging data that can be used for precise analysis. To date, even in view of the rapid progress of voltage-sensitive indicators and relevant imaging technologies in recent years, this very old 'art' of combining single-cell electrophysiology and two-photon imaging (ordinary, raster-scanned, video-rate imaging) of Ca2+ signals still enables measurements of the best level precision.

We thank the reviewer for reminding us of these important previous studies that we cite now in the revised manuscript.

On the other hand, the interpretation of data in this study is a bit narrow-minded and lacks a comprehensive picture. Some suggestions to improve the manuscript are as follows:(1) The authors made a segregation of 'spine synapse' and 'shaft synapse' based solely on the two photon images in-vivo. However, caution shall be taken here, because the optical resolution under in vivo imaging conditions like this cannot reliably tell apart whether a bright spot within or partially overlapping a segment of the dendrite is a spine on top of (or below) it. Therefore, what the authors consider as a 'shaft synapse' (by detecting Ca2+ hotspots) has an unknown probability of being just a spine on top or below the dendrite. If there is other imaging data of higher axial resolution to validate or calibrate, the authors shall take some further considerations or analysis to check the consistency of their data, as the authors do need such a segregation between spine and shaft synapses to show how they evolve over the brain development stages.

We agree with the reviewer that the differentiation between spine and sha synapses can be difficult for those spines that are located above or below the dendric sha in the z-dimension because of the lower resolution of 2-photon microscopy in the z-dimension compared to the image plane. We have now added a new paragraph to the Methods section to describe in more detail how we identify spine and sha synapses and provide more examples in a new supplementary figure (Fig S5). We believe that we can identify spine and sha synapses reliably in most cases, but added a cautionary note to make the reader aware of potential misidentifications.

(2) The use of terminology 'bursts of spontaneous inputs' for describing voltage-clamp data seems improper. Conventionally, 'burst' refers to suprathreshold spike firing events, but here, the authors use 'burst' to refer to inward synaptic currents collected at the cell body. Not every excitatory synaptic input (or ensemble of inputs) activation will lead to spike firing under naturalistic conditions, therefore, these two concepts are not equivalent. It is recommended to use 'barrage of inputs' instead of 'burst of inputs'. Imagine a full picture of the entire dendritic tree, the fact that the authors could always capture spontaneous Ca2+ events here and there within a few pieces of dendrites within an arbitrary field-of-view suggests that, the whole dendritic tree must have many more such events going on as a barrage while the author's patch electrode picks up the summed current flow from the whole dendritic tree.

We agree with the reviewer that “barrage” is a clearer term for multiple synaptic inputs occurring simultaneously and therefore we changed the terminology throughout the manuscript.

(3) Following the above issue, an analysis of the temporal correlation between synaptic (not segregating 'spine' or 'shaft') Ca2+ events and EPSCs is absent. Again, the authors drew arbitrary time windows to clump the events for statistical analysis. However, the demonstrated example data already shows that the onset times of individual synaptic Ca2+ events do not necessarily align with the beginning of a 'barrage' inward current event.

The reviewer writes that “an analysis of the temporal correlation between synaptic calcium events and EPSCs is absent”. We would like to point out that we did determine the percentage of calcium transients that occurred during barrages of synaptic inputs (~60%, page 7). This is important, since the barrages in our patch-clamp recordings most likely reflect spontaneous network events as described in the developing cortex previously by us and many other labs . The time window we chose was not “arbitrary” as the reviewer suggests, but based on the duration of the barrages of synaptic inputs as defined in the Methods section.

The reason, why we did not perform a more in-depth analysis of the temporal relationship between synaptic calcium transients and synaptic input currents is that it is essentially impossible to relate calcium transients at individual synapses to specific synaptic input events. First, during barrages of synaptic inputs many synapses are active simultaneously, both in the mapped dendrites as well as in the un-observed parts of the dendric arborization as the reviewer notes above. Thus, barrages cannot be broken down into individual synaptic transmission events. Second, since our acquisition frequency is ~10 Hz, we can identify the onset of individual synaptic calcium transients with 100-200 ms precision (1 or 2 frames). However, throughout any 100-200 ms period of recording, several synapses are active across the entire dendric arborization such that we cannot assign a given calcium transient to a specific EPSC within a 100-200 ms epoch. Third, due to the limited clamping capacity of in vivo patch recordings, we cannot be certain that individual transmission events in distal dendrites can be resolved in the patch recording.

(4) The authors claim that "these observations indicate that the activity patterns investigated here are not or only slightly affected by low-level anesthesia". It would be nice to show some of the recordings in this work without any anesthesia to support this claim.

Indeed, the conclusion that the patterns of activity are only slightly affected by low levels of anesthesia is based on our previous recordings on the network level. Unfortunately, we are still not able to record calcium imaging with single synapse resolution in unanesthezed developing mice (and no one else is as far as we know), because the skull of these young animals is not firm, yet. As a consequence, movements cannot be reduced sufficiently for patching and imaging with single synapse resolution. Our previously published (Siegel et al., 2012) and unpublished work on the cellular level suggests that activity patterns during light anesthesia are very similar to those during sleep in mouse pups at this age.

**Reviewer #3 (Public Review)**:Summary:There is a growing body of litterature on the clustering of co-active synapses in adult mice, which has important implications for understanding dendritic integration and sensory processing more broadly. However, it has been unclear when this spatial organization of co-active synapses arises during development. In this manuscript, Leighton et al. investigate the emergence of spatially organized, coactive synapses on pyramidal dendrites in the mouse visual cortex before eye-opening. They find that some dendrite segments contain highly active synapses that are co-active with their neighbors as early as postnatal day (P) 8-10, and that these domains of co-active synapses increase their coverage of the dendritic arbor by P12-13. Interestingly, Leighton et al. demonstrate that synapses co-active with their neighbors are more likely to increase their activity across a single recording session, compared to synapses that are not co-active with their neighbors, suggesting local plasticity driven by coincident activity before eye-opening.The current manuscript includes some replication of earlier results from the same research group (Winnubst et al., 2015), including the presence of clustered, co-active synapses in the visual cortex of mouse pups, and the finding that synapses co-active with their neighbors show an increase in transmission frequency during a recording session. The main novelty in the current study compared to Winnubst et al. (2015) is the inclusion of younger animals (P8-13 in the current study compared to P10-15 in Winnubst et al., 2015). The current manuscript is the first demonstration that active synapses are clustered on specific dendrite segments as early as P8-10 in the mouse visual cortex, and the first to show the progression in active synapse distribution along the dendrite during the 2nd postnatal week. These results from the visual cortex may help inform our understanding of sensory development more broadly.Strengths:The authors ask a novel question about the emergence of synaptic spatial organization, and they use well-chosen techniques that directly address their questions despite the challenging nature of these techniques. To capture both structural and functional information from dendrites simultaneously, the authors performed a whole-cell voltage clamp to record synaptic currents arriving at the soma while imaging calcium influx at individual synaptic sites on dendrites. The simultaneous voltage clamp and calcium imaging allowed the authors to isolate individual synaptic inputs without their occlusion by widespread calcium influx from back-propagating action potentials. Achieving in vivo dendrite imaging in live mice that are as young as P8 is challenging, and the resulting data provides a unique view of synaptic activity along individual dendrites in the visual cortex at an early stage in development that is otherwise difficult to assess.The authors provide convincing evidence that synapses are more likely to be co-active with their neighbors compared to synapses located farther away (Fig. 6F-H), and that synapses co-active with their neighbors increase their transmission frequency during a recording session (Figure 7C). These findings are particularly interesting given that the recordings occur before eye-opening, suggesting a relationship between co-activity and local synaptic plasticity even before the onset of detailed visual input. These results replicate previously published findings from P10-15 pups (Winnubst et al., 2015), increasing confidence in the reproducibility of the data.The authors also provide novel data documenting for the first time spatially organized, co-active synapses in pups as young as P8. Comparing the younger (P8-10) and older (P12-13) pups, provides insight into how clusters of co-active synapses might emerge during development.Weaknesses:This manuscript provides insufficient detail for assessing the rigor and reproducibility of the methods, particularly for age comparisons. The P8-10 vs P12-13 age comparisons are the primary novel finding in this manuscript, and it is, therefore, critical to avoid systematic age differences in the methods and analysis whenever possible. Specific concerns related to the age comparisons are listed below:(1) Given that the same research group previously published P12-13 data (Winnubst et al., 2015), it is unclear whether both age groups in the current study were imaged/analyzed in parallel by the same researcher(s), or whether previous data was used for the P12-13 group.

While indeed the approach in the present study is similar to that of our previous study (Winnubst et al. 2015), the data set presented here is entirely new. The current study was made possible by a new microscope that allows combining resonant scanning with piezo-focusing to image large fractions of the dendric arborization. In fact, we could now image almost 10 times larger dendric segments including branch points than in our previous study. One author contributed to the experiments in both studies. Image analysis of all experiments was performed by the first author of the present study who was not involved in the Winnubst et al. work.

(2) The authors mention that they used 2 different microscopes, and used a fairly wide range of imaging frame rates (5-15 Hz). It is unclear from the current manuscript whether the same imaging parameters were used across the two age groups. If data for the two experimental groups was collected separately, perhaps at different times, by a different person, or on a different microscope, there is a concern that some differences between the groups may not necessarily be due to age.

The reviewer mentions that the experimental settings are not identical across the experiments of this study. In the original manuscript we erroneously reported in the Methods section that 2 different setups were used for this study; however, all experiments were performed on the same microscope. We have corrected this in the new manuscript. We took timelapse recordings of small stacks of varying depth to cover as many dendrites as possible in each recording, therefore, we needed to adjust the rate of acquired stacks within a certain range as the reviewer points out. The data were acquired by two scientists during an overlapping period. And while the different ages were not recorded in a strictly randomized fashion, they were not acquired in sequence according to ages, but rather involved many attempts on animals of different ages from many different litters. For each litter a small percentage of animals would generate successful recordings, and the ages of these successes were random. Therefore, we believe that neither the collection of data nor the analysis (see point above) affected the differences we describe here for the two age groups.

(3) It is unclear whether the image analysis was performed blind to age. Blinding to age during analysis is particularly important for this study, in which it was not possible to blind to age during imaging due to visible differences in size and developmental stage between younger and older pups.

The analysis was not setup to be performed blind to age. Not only is the age of the animal apparent at the stage (as the reviewer points out), also the number of spines and the activity levels clearly show differences between neurons only a few days apart. However, all age-related findings reported in this study - except the increase in synapse density and activity - became apparent to us only after the full set of synaptic transmission events was determined and the analysis was performed on the entire data set, making it very unlikely that event detection was biased.

(4) The relatively low N (where N is the number of dendrites or the number of mice) in this study is acceptable due to the challenging nature of the techniques used, but unintentional sampling bias is a concern. For example, if higher-order dendrites from the apical tuft were imaged at P12-13, while more segments of the apical trunk were imaged at P8-10, this could inadvertently create apparent age differences that were in fact due to dendrite location on the arbor or dendrite depth.

The reviewer points out that sampling bias with respect to synapse location along dendrites in the dataset could lead to falsely apparent age differences. In all experiments we imaged dendrites of layer 2/3 neurons that were relatively close to the cortical surface to optimize image quality. In addition, we confirmed that the mean distance of the imaged dendric stretches from the cell body was similar between the dendrites of each age group (Young: 392 +/- 104 µm, Old: 323 +/- 118 µm; mean +/- STD). Therefore, we do not think that sampling bias affected these results.

Additional general methodological concerns, which are not specifically related to the age comparisons, are listed below:(5) The authors assert that clustered, co-active synapses emerge in the visual cortex before eye-opening, which is an important finding in that it suggests this phenomention is driven by spontaneous activity rather than visual input. However, this finding hinges on the imaged cells being reliably located in the visual cortex, which is difficult to identify with certainty in animals that have not yet opened their eyes and therefore cannot undergo intrinsic signal imaging to demarcate the boundaries of the visual cortex. If the imaged cells were in, for example, nearby somatosensory cortex, then the observed spatial organization could be due to sensory input rather than spontaneous activity.

The reviewer argues that if the neurons included in our analysis were located in non-visual sensory cortex, e.g. the somatosensory cortex, sensory experience might have shaped clustered inputs instead of spontaneous activity. We are, however, certain that the neurons were located inside the primary visual cortex. In previous experiments where we performed the same craniotomies, we mapped spontaneous activity across the sensory areas in the occipital neocortex and we know the exact location of V1 which is already very consistent during the second postnatal week. (See for example Supplemental Figure 4 in Leighton et al., 2021).

(6) It is unclear how the authors defined a synaptic transmission event in the GCaMP signal (e.g. whether there was a quantitative deltaF/F threshold).

In the revised manuscript, we describe the procedure of identifying synaptic calcium transients in more detail and added a new supplemental figure to clarify this aspect of the analysis. In short, we use an automated detection with a 2x standard deviation threshold and a subsequent manual control and selection step. Please, find all details in the Methods section and Figure S4 of the revised manuscript.

(7) The authors' division of synapses into spine vs shaft is unconvincing due to the difficulty of identifying Z-projecting spines in images from 2-photon microscopy, where the Z resolution is insufficient to definitively identify Z-projecting spines, and the fact that spines in young animals may be thin and dim. The authors' examples of spine synapses (e.g. in Fig. 2A) are convincing, but some of the putative shaft synapses may in fact be on spines.

We agree with the reviewer that the differentiation between spine and sha synapses can be difficult for those spines that are located above or below the dendric sha in the z-dimension because of the lower resolution of 2-photon microscopy in the z-dimension compared to the image plane (see also response to Reviewer 2, point 1). We have now added a new paragraph to the Methods section to describe in more detail how we identify spine and sha synapses and provide more examples in a new supplementary figure (Fig S5). We believe that we can identify spine and sha synapses reliably in most cases, but added a cautionary note to make the reader aware of potential misidentifications.

**Reviewer #1 (Recommendations For The Authors):**
I think the experiments performed were very technically challenging (probably one of the few labs that can do this in the field), and the findings provide in vivo evidence on how structured synaptic inputs are assembled during development that has never been reported.I suggest improving the writing and presentation and really explaining how they conducted the experiments and how they defined shaft synapses.Line 96: 12 dendritic areas from 11 mice at ages between postnatal day 8 to 13.- Do the authors know how many neurons were imaged? It is unclear if the authors patch on all the imaged neurons and only imaged (or analyzed) the dendrites of those patched neurons. If yes, how sparse are the neurons labelled from IUE? From 1B, it looks like there are two cells adjacent to each other. Can the authors really distinguish whether the imaged dendrites are from the patched neuron?

The reviewer wonders whether we can tell apart dendrites of patched cells from those of neighboring neurons that were not patched. This is actually very straight forward: the experiment included a depolarization step (see Methods section) which leads to an immediate, but temporary, increase in fluorescence in all of the patched neurons’ dendrites, but none of the neighboring dendrites. We have added this information to the Methods section of the new manuscript and provide now an example (Fig S3). Furthermore, as these cells normally fire frequently, it would immediately become clear that an unpatched cell is being imaged if backpropagating action potentials are predominantly observed rather than synaptic signals. The visualization of these synaptic signals is only possible due to the blockade of Na+ channels with QX314 in the intracellular solution (see Methods).

- In the methods section, it says 'dendrites were imaged in single plane or small stacks with plane...'. How do the authors do calcium imaging with small stacks of plane using Nikon MP scope?

Small stacks were acquired by using the piezo focusing device of our Nikon A1 microscope. Since we combined this fast focusing approach with resonant scanning, we were able to acquire z-stacks of 3-5 frames at a rate of up to 15 Hz (per stack).

- I also assume this is not chronic imaging, and there are different mice for each postnatal day. If it's true, this is somewhat important for all the correlation analysis as there are only 2 mice for each postnatal day (other than day 12) and day 13 only has 1 animal.

Yes, indeed these are not chronic experiments and dendrites imaged on different days are from different neurons and different mice. We agree with the reviewer that if it had been possible to image the same neurons across these developmental stages, we would have detected even clearer correlations. Therefore, we see our results as conservative estimates of the developmental trajectory of the analyzed parameters.

Line 104 - 109: I don't understand why the authors need to hold at -30mV to facilitate calcium influx through NMDA receptors? I assume this helps them to visualize as many synapses as possible? but wouldn't that also make the 'event frequency' not reflect the true value?

Indeed depolarizing the imaged neurons to -30 mV was necessary to get sufficient calcium influx to map synaptic inputs. We don’t think that this affects the frequency of inputs, because the frequency of synaptic inputs is determined by the presynaptic firing rate and the release probability of the presynaptic terminal, which are not affected by the depolarization of the dendrite.

Figure 2A - It says in the method section that ROIs are manually selected. However, it's not explained what the criteria are. For spine synapses, it's easy to define but for shaft synapses like in Fig 2B, why are there 2 synapses on the shaft? And in Fig 4a, 5a, Fig S1 P13, some of the dendrites are packed with ROIs. What's the distance between those shaft synapses? Can the imaging resolution really separate them?

The reviewer asks for a better description of how we identified individual ROIs and thus synapse locations and whether this is actually feasible. We have now added a more detailed description of how we select synaptic sites based on the occurrence of synaptic calcium transients. In addition, we have added a new supplemental Figure (S4) to give the reader an impression of the image quality and the ability to locate individual synapses reliably. We find that separating sha synapses was possible for inter-synapse distances of ~4 µm or more. The mean sha synapse distance in our data set is 21 µm.

- Similar issue applies to Figure 4A that I'm not sure what's the resolution of each 'hot spot'. They all seem very close together. Maybe additional raw dendrite images with fluorescence changes like 1C or 2A could be helpful (or movies in the supplementary?)

As the reviewer suggests, we have added now additional supplemental figures to illustrate better how we identify synaptic transmission events as well as spine and sha synapses.

- Also for line 164, it says that 76% of high-activity synapses were located on spines. This could also maybe support that only the spine synapses are real synapses and many shaft synapses are actually not synapses and they were just categorized as shaft synapses from manual ROI?

We are actually quite sure that sha synapses are real synapses based on our analysis, since they show repeated synaptic calcium transients that co-occur with barrages of synaptic inputs as measured by patch-clamp recordings. Indeed one would expect to see a number of excitatory synapses on dendric shas of pyramidal neurons at these ages based on previous EM studies (Miller and Peters, 1981; Wildenberg et al., 2023).

- While this might not impact the overall novelty of the paper, I would be curious to know if the authors can still observe the same findings if they only analyze spine synapses.

We repeated several analyses with a dataset that contained only spine synapses. For most analyses we observed the expected result: the effect sizes were similar compared to the entire data set, but the power was reduced. For example the effect of distance to closest high-activity neighbor and own activity (Fig 5E, F) was similar, but p-values were around 0.1 (Similar results for Figure 7B). In contrast, the co-activity with synapses within a domain was significantly higher than the co-activity with synapses in other domains also for the spine-synapse only data set.

Fig 6 - Does the domain co-activity also contribute to the synaptic current recorded (related to Fig 4).

Yes, the synaptic activity measured by calcium imaging contributes to the recorded EPSCs. However, the exact relationship between synaptic inputs measured by calcium imaging and those measured by patch-clamping is complicated by 3 factors: first, during barrages of synaptic inputs many synapses are active simultaneously, both in the mapped dendrites as well as in the un-observed parts of the dendric arborization. Thus, barrages cannot be broken down into individual events. Second, since our acquisition frequency is ~10 Hz, we can identify the onset of individual synaptic calcium transients with 100-200 ms precision (1 or 2 frames). However, throughout any 100-200 ms period of recording several synapses are active across the entire dendric arborization such that we cannot assign a given calcium transient to a specific EPSC within a 100-200 ms epoch. Third, due to the limited clamping capacity of in vivo patch recordings, we cannot be certain that individual transmission events in distal dendrites can be resolved in the patch recording as EPSCs.

**Reviewer #2 (Recommendations For The Authors):**
(1) I suggest the authors should provide the number of cells and mice recorded in the figure legends.

The number of dendrites and mice is the same across all analyses: 12 dendrites from 11 mice for all experiments, 6/6 for P8-10 and 6/5 for P12-13. All dendrites and synapses (and their ages) are shown in the supplemental figures S1 and S2. We mention the number of imaged dendrites now at the beginning of the Results section and when we split ages for the first me.

(2) Instead of showing only cartoon illustrations of dendrites in Figures 3-6, I suggest showing the two photon images as well together with the cartoon.

The 2-photon images of all dendrites of the dataset are available in Figure S1. To allow the reader to compare the cartoon representations in the main figures and the 2-photon images of each neuron, we have now labeled each dendrite in the dataset (D1-D12, see figures S1 and S2). For every figure, where we show example neurons (cartoons or zoom ins) we now provide this identifier.

**Reviewer #3 (Recommendations For The Authors):**
To address the weaknesses outlined above, we recommend that the authors do the following:• To address concerns about the rigor and reproducibility of the methods specifically related to age comparisons, please confirm the following:- Both age groups were run in parallel by the same researcher(s).

Experiments were run partly overlapping and experiments from different age groups were performed in parallel by both researchers.

- Both age groups were imaged on the same microscope, or animals from each age group were imaged on both microscopes. If it was necessary to use different microscopes for the different age groups for biological or practical reasons, please explain.

All experiments were run on the same microscope, a Nikon A1 2-photon microscope. In the original methods description we erroneously mentioned two microscopes (copy and paste error from a previous publication). We corrected that in the revised manuscript.

- There was no difference in imaging frame rates or other imaging parameters between age groups. If it was necessary to use different parameters for different age groups for biological reasons, please explain.

We varied the frame rates somewhat to allow larger z-stacks for some experiments where dendrites traversed different depths; however the mean frame rates were similar between the experiments in P8-10 vs P12-13 dendrites, 8.5 vs 10 Hz, respectively.

- Images were analyzed blind to age.

The analysis was not setup to be performed blind to age. The number of spines and the activity levels clearly show obvious differences between neurons only a few days apart. However, all findings reported in this study related to age - except the increase in synapse density and activity - became apparent to us only after the full set of synaptic transmission events was determined and the analysis was performed on the entire data set, making it unlikely that event detection was biased.

- There was no difference in the location of analyzed dendrites (e.g. depth from the pia, branch order) between age groups.

In all experiments we imaged dendrites of layer 2/3 neurons that were relatively close to the cortical surface to optimize image quality. In addition, we determined the mean distance of the imaged dendric stretches from the cell body and found that this distance was similar between the dendrites of each age group (Young: 392 +/- 104 µm, Old: 323 +/- 118 µm; mean +/- STD). Therefore, we do not think that sampling bias affected these results.

• To address general methodological concerns, please provide additional description of the following points:- Please clarify how the visual cortex was identified in P8-13 pups. If there was ambiguity about identifying the visual cortex in these pups, please discuss the implications of this ambiguity.

The reviewer asks how we identified V1 in these experiments. We are indeed certain that the neurons were located inside the primary visual cortex. We have ample experience with mapping V1 in these animals based on patterns of spontaneous activity as well as post-hoc stainings. V1 is quite large already at these ages (> 2 mm long and > 1 mm wide) and its extent very consistent across animals. Thus, we would argue it is actually hard to miss.

- Please clarify how synaptic transmission events were identified in the GCaMP signal.

We have now added a more detailed description of how we identify synaptic calcium transients. In addition, we have added a new supplemental Figure (S3) to give the reader an impression of the image quality and the ability to locate individual synapses reliably.

- It is acceptable to use the spine vs shaft analysis despite the inevitable difficulty resolving Z-projecting spines, but this caveat should be mentioned in the discussion of the spine vs shaft results.

We added a more detailed description of spine and sha synapse identification, a new supplemental figure (S5) and we now mention the caveat related to the limited z-resolution of 2-photon microscopy in the revised manuscript.

• Two additional minor details should be clarified in the text of the manuscript:- Please specify the volume of DNA solution injected into each embryo.

The injected volume was 1 µl. We added this information in the Methods section of the revised manuscript.

- In Fig S1, please specify whether the scale bar applies to all images.

The scale bar applies to all images. This information was added to the figure legend.

References

Leighton AH, Cheyne JE, Houwen GJ, Maldonado PP, De Winter F, Levelt CN, Lohmann C. 2021. Somatostatin interneurons restrict cell recruitment to renally driven spontaneous activity in the developing cortex. Cell Rep 36:109316. doi:10.1016/j.celrep.2021.109316

Miller M, Peters A. 1981. Maturation of rat visual cortex. II. A combined Golgi-electron microscope study of pyramidal neurons. JComp Neurol 203:555–573.

Siegel F, Heimel JA, Peters J, Lohmann C. 2012. Peripheral and central inputs shape network dynamics in the developing visual cortex in vivo. Current Biology 22:253–258.

Wildenberg G, Li H, Sampathkumar V, Sorokina A, Kasthuri N. 2023. Isochronic development of cortical synapses in primates and mice. Nat Commun 14:8018. doi:10.1038/s41467-02343088-3